# Preparation and Characteristics of Polyethylene Oxide/Curdlan Nanofiber Films by Electrospinning for Biomedical Applications

**DOI:** 10.3390/ma16103863

**Published:** 2023-05-20

**Authors:** Shu-Hung Lin, Sin-Liang Ou, Hung-Ming Hsu, Jane-Yii Wu

**Affiliations:** 1PhD Program of Biotechnology and Industry, College of Biotechnology and Bioresources, Da-Yeh University, Changhua 515, Taiwan; shuhunglin1969@gmail.com; 2Department of Biomedical Engineering, Da-Yeh University, Changhua 515, Taiwan; 3Department of Medicinal Botanicals and Foods on Health Applications, Da-Yeh University, Changhua 515, Taiwan; a0933571498@gmail.com; 4Biotechnology Research and Development Center, Da-Yeh University, Changhua 515, Taiwan; 5Innovation Incubation Center, Da-Yeh University, Changhua 515, Taiwan

**Keywords:** electrospinning, polyethylene oxide, curdlan, nanofiber films, wetting time, disintegration time

## Abstract

In this study, polyethylene oxide (PEO) and curdlan solutions were used to prepare PEO/curdlan nanofiber films by electrospinning using deionized water as the solvent. In the electrospinning process, PEO was used as the base material, and its concentration was fixed at 6.0 wt.%. Moreover, the concentration of curdlan gum varied from 1.0 to 5.0 wt.%. For the electrospinning conditions, various operating voltages (12–24 kV), working distances (12–20 cm) and feeding rates of polymer solution (5–50 μL/min) were also modified. Based on the experimental results, the optimum concentration for the curdlan gum was 2.0 wt.%. Additionally, the most suitable operating voltage, working distance and feeding rate for the electrospinning process were 19 kV, 20 cm and 9 μL/min, respectively, which can help to prepare relatively thinner PEO/curdlan nanofibers with higher mesh porosity and without the formation of beaded nanofibers. Finally, the PEO/curdlan nanofiber instant films containing 5.0 wt.% quercetin inclusion complex were used to perform wetting and disintegration processes. It was found that the instant film can be dissolved significantly on the low-moisture wet wipe. On the other hand, when the instant film touched water, it can be disintegrated very quickly within 5 s, and the quercetin inclusion complex was dissolved in water efficiently. Furthermore, when the instant film encountered the water vapor at 50 °C, it almost completely disintegrated after immersion for 30 min. The results indicate that the electrospun PEO/curdlan nanofiber film is highly feasible for biomedical applications consisting of instant masks and quick-release wound dressings, even in the water vapor environment.

## 1. Introduction

The nanofiber films produced by electrospinning have characteristics such as softness, high strength, high bulkiness and good filtering effect. Therefore, these nanofiber films can be used in related filter materials consisting of air filters, air conditioner filters, antibacterial dust masks, and so on. In addition, for medical applications, the electrospun nanofiber films have several advantages consisting of a larger specific surface area, good ductility, high elasticity, adjustable porosity, excellent moisture retention, absorption and compliance [1,2]. At the same time, it can also carry drugs or other functional ingredients. Thus, it can be employed as a wound dressing and added with biologically active substances such as growth factors and antibacterial ingredients to prevent wound infection. Due to its biodegradable characteristics, it can be gradually disintegrated during the wound repair process, avoiding the disadvantages of traditional gauze that would cause discomfort to the affected part during the removal process. 

On the other hand, in the application of biomedical products, this type of film can be used in cosmetic masks. It is expected that the cosmetic mask prepared with the nanofiber film will closely adhere to the texture of the skin, and have a high moisturizing effect to help the skin lock in moisture strongly. Moreover, because of its high compliance, it can produce a sealing effect on the skin, forcing the pores to open. Then, through the close fit of dressing and skin, the temperature of the local skin can rise, increasing the skin’s absorption of ingredients. When the skin temperature rises and the pores open, the composition of the mask itself will start to dissolve, and the essential components will slowly separate out of the mask and penetrate into the skin. In comparison to traditional masks, the essential components of the highly biocompatible nanofiber masks are relatively difficult to evaporate and dry out. At the same time, due to its dissolvable properties, it can effectively reduce the subsequent environmental protection treatment issues derived from waste masks. 

Electrospinning is a technique that can easily process polymers into nanofibers. The nanofiber films can be fabricated by electrospinning with many biocompatible polymer materials, such as hyaluronic acid (HA), gelatin, polyglutamic acid (PGA), polyvinyl alcohol (PVA), polyethylene oxide (PEO), curdlan gum, etc. HA is widely investigated because of its high potential for wound dressing applications [3]. The fabrication of biomimetic HA-based scaffolds by electrospinning is thus extensively investigated [4]. However, electrospun HA with PVA as a carrier polymer allowed the processing in water but with poor efficiency and the scaffolds showed many defects [4]. Gelatin is a natural polymer derived from collagen by controlling hydrolysis, which has a lot of integrin-binding sites for adhesion and differentiation [5]. However, pure gelatin nanofibrous scaffolds are too fragile to be handled in practical applications [6]. PGA is a natural polymer (often produced by the fermentation of bacteria) that is originally used for wound healing. Moreover, the electrospun PGA polymer can be employed as a bone tissue construct [7]. PVA, which is essentially produced from polyvinyl acetate through hydrolysis, is easily degradable by biological organisms [8]. PVA has been employed to produce many products including lacquers, resins, surgical threads, and food packaging materials. The electrospinning of the PVA solution has been widely investigated for the preparation of ultrafine separation filters, biodegradable mats and inorganic fibers [9]. Among these polymer materials, PEO and curdlan gum have quite good properties as raw materials for electrospinning. The chemical structure of PEO is commonly expressed as H−(O−CH_2_−CH_2_)_n_−OH. PEO has high hydrophilicity, good biocompatibility, good mechanical properties, low toxicity, excellent spinnability, high molecular weight and viscosity [10]. When other polymer materials are mixed and co-electrospun with PEO, PEO can significantly promote the absence of entangled and interacting polymers during the electrospinning process and has the advantages of good thermal stability and excellent porosity [11,12]. Besides, curdlan is a neutral microbial polysaccharide with a linear structure composed of D-glucose chains formed from β-1, 3 glucoside bonds produced by *Agrobacterium*, *Rhizobium* and *Alcaligenes faecalis* [13]. It is a natural polysaccharide and is hardly soluble in water and alcohols. This kind of colloid can be dispersed in water and gelled under heating and alkaline conditions. Because of this unique gelling property, it is widely used in the food industry as an emulsifier, processing aid, stabilizer, thickener and quality improver. It is known that this colloid has anti-tumor, anti-HIV, anti-inflammatory, anti-infection, immune activity promotion and wound repair properties [14]. 

The electrospinning of curdlan alone is difficult due to its gelation, which takes place at a relatively low polymer concentration (5 wt.% and above) [15], while at low concentrations the curdlan solution does not produce fibrous structures. Thus, a successful electrospinning process can be obtained only in the curdlan blended with other polymers. PEO, a biocompatible, cost effective and non-toxic hydrophilic polyether [16], is a suitable choice for co-electrospinning with curdlan to make nanofiber films. So far, research on the co-electrospinning of PEO and curdlan to prepare PEO/curdlan nanofiber films is still quite rare. El-Naggar et al. successfully prepared PEO/curdlan nanofiber films using co-electrospinning [15], and the presence of defect-free smooth fibers can be achieved using 6% PEO and the PEO/curdlan solution with a 60/40 weight ratio. However, based on the release of tetracycline hydrochloride from electrospun webs of the PEO/curdlan (60/40) blend revealed that the electrospun nanofiber film provided relatively slow drug release over about 24 h. 

In this study, PEO was used as the base material mixed with cardlan gum, and electrospinning was performed to prepare PEO/curdlan nanofiber films. After adding an appropriate amount of curdlan gum, the viscosity and conductivity of the solution can be improved [17]. The increase in the conductivity of the solution can make the solution easier to be stretched during the electrospinning process, and obtain a finer and smoother fiber filament. If the viscosity of the solution is too low, the spherical protrusions such as beads are easily formed in the filaments during the electrospinning process. However, as the viscosity of the solution is increased, the protrusions gradually shrink from spherical to conical, and the linear fibers can be further formed. The influence of operating parameters (operating voltage, working distance and feeding rate of polymer solution) in the electrospinning process on the films’ properties was discussed. Furthermore, the wetting and disintegration processes were also performed on the PEO/curdlan nanofiber film to estimate its feasibility for biomedical applications. Although PEO/curdlan nanofiber films were successfully fabricated by electrospinning in previous research [15], the adjustment of the process focused on the changes in PEO concentration and the ratio of PEO/curdlan in the solution. In our research, the concentration of curdlan was changed. More importantly, we carefully adjusted the process parameters of electrospinning. This has not been explored in previous studies on the fabrication of electrospun PEO/curdlan nanofiber films. In this way, the influence of electrospinning parameters on the properties of PEO/curdlan nanofiber films will be fully understood. 

## 2. Materials and Methods

In this research, PEO powders with an average molecular weight of 100,000 g/mol were purchased from Scientific Polymer Product Company, Ontario, NY, USA. Moreover, curdlan (Sigma Aldrich, St. Louis, MO, USA) was produced from *Alcaligenes faecalis* and tetracycline hydrochloride. PEO and curdlan solutions were used to prepare PEO/curdlan nanofiber films by electrospinning, and deionized water was employed as the solvent. All chemicals were of analytical grade and were used without further purification. The solution was then stirred overnight at room temperature using a magnetic stirring plate to ensure a homogenous solution. Additionally, for the wetting and disintegration measurements, the 5.0 wt.% quercetin inclusion complex (Sigma Aldrich, St. Louis, MO, USA) was also encapsulated in the PEO/curdlan nanofiber films. Quercetin (C_15_H_10_O_7_-3,3′,4′,5,7-pentahydroxyfavone), a polyhydroxy compound and a natural flavonoid substance, has attracted much attention in recent years because of its beneficial properties to human health. 

According to the measurement results, when the concentration of PEO and the volume ratio of PEO/curdlan gum were 6.0 wt.% and 90/10, respectively, the PEO/curdlan nanofiber film had the highest swelling degree. Thus, in this research, the concentration of PEO and the volume ratio of PEO/curdlan gum were kept at 6.0 wt.% and 90/10, respectively. First, a sufficient amount of PEO was put in deionized water and then heated to dissolve it, so that the final concentration is 6.0 wt.%. After stirring 6.0 wt.% PEO solution with various concentrations (1.0, 2.0 and 5.0 wt.%) of curdlan gum in a fixed volume ratio of 90:10 was obtained; the mixed solution can be used to prepare nanofiber films by electrospinning. Moreover, for wetting and disintegration measurements, 5.0 wt.% quercetin was added to the PEO/curdlan solution and mixed for 4 h at 40 °C to obtain a homogenous solution. 

The basic configuration of an electrospinning device includes (1) a capillary tube with an injection needle or pipette; (2) a high-voltage current supplier and (3) a collector or collecting plate. The high-voltage current supply, the capillary tube containing the polymer solution and the collecting plate were connected by using a wire. The distance between the capillary tube and the collecting plate must be kept short. A metal plate or aluminum foil can be used as a collecting plate to collect filaments. In the electrospinning process, the system was adjusted at room temperature first, and the mixed solution was poured into a 10 mL syringe. Then it was put in the injection pump, which slowly pushed the polymer solution into the spinneret by the thrust of the pump. The needle of the syringe was connected to a high-voltage power supply so that the front of the spinneret was negatively charged and the collector plate was grounded to be positively charged. A voltage is applied to charge the surface of the droplet to form an electrostatic repulsion sufficient to overcome the surface tension, allowing the droplet to form a Taylor cone, thereby stretching the spinneret. Finally, the fibers are collected on stainless steel metal drums. During the electrospinning process, various operating voltages (12–24 kV), working distances (12–20 cm) and feeding rates of polymer solution (5–50 μL/min) were also modified to obtain the high-quality PEO/curdlan nanofiber films. The working distance means the distance between the tip of the capillary tube and the collecting plate. According to our experimental results, the nanofiber films produced by changing these parameters (operating voltage, working distance and feeding rate) with the above ranges have more obvious mesh-porous structures, and these parameters are also within the allowable operating range of our experimental equipment. To analyze the wetting and disintegration times of PEO/curdlan nanofiber films, the films were put in the wet wipe, water and water vapor for observation. The morphologies of electrospun nanofiber films were observed by scanning electron microscopy (SEM, Hitachi SU 8200, Tokyo, Japan). The average fiber diameter was determined by measuring 100 fibers selected randomly from each sample. The mesh porosity of the nanofiber film was determined using a gravimetric method based on the following relation:P(%) = (1 − d_s_/d_p_) × 100%
where d_s_ and d_p_ are sample density and polymer density, respectively. Square samples with dimensions about 10 mm × 10 mm were weighed, and sample density were determined by d_s_ = m/(δ·A), where m, δ and A are sample mass (g), sample thickness (cm) and sample area (cm^2^), respectively [18]. The sample thickness was measured by SEM. The measurement of average mesh porosity for each sample was performed three times. 

## 3. Results and Discussion 

### 3.1. Effect of Operating Voltage on the Formation of Nanofiber Films

Appendix A shows SEM images and fiber-diameter scatter diagrams of PEO/curdlan nanofiber films prepared by electrospinning under various applied voltages (12, 14, 17.5, 19, 22 and 24 kV). The polymer solution used in the electrospinning process was mixed with 6.0 wt.% PEO and 1.0 wt.% cardlan gum. Here, the working distance and feeding rate were fixed at 15 cm and 9 μL/min, respectively. From our observation, when the applied voltage was between 12 and 17.5 kV, many bead-shaped nanofibers can be clearly found and the diameter of fiber filaments was generally finer. As the voltage was gradually increased, the nanofiber diameter also gradually increased. However, there was a gradual decrease in the mesh porosity, as shown in Figure 1. 

Appendix A shows SEM images and fiber-diameter scatter diagrams of PEO/curdlan nanofiber films prepared by electrospinning under various DC voltages (12, 14, 16, 17, 17.5, 19, 22 and 24 kV). The polymer solution used in the electrospinning process was mixed with 6.0 wt.% PEO and 2.0 wt.% cardlan gum. As the operating voltage was 12 kV, the fiber diameter reached 302 nm, and the mesh porosity was 0.71%. However, when the voltage was increased to 14 kV, the fiber diameter dropped sharply to 103.3 nm, and the mesh porosity increased to 5.07%. With increasing the applied voltage to 14–17.5 kV, the fiber diameter was maintained in the range of 103–109 nm, and the mesh porosity was between 5.07% and 11.16%. When the applied voltage was increased to 19 kV, the fiber diameter decreased to 93.5 nm, and the mesh porosity increased to 13.37%. Additionally, the fiber filaments were more uniform. At the same time, as the voltage gradually increased, the fiber diameter can continuously reduce, while the mesh porosity increased. When the applied voltage was 24 kV, the lowest fiber diameter of 54.2 nm and the highest mesh porosity of 21.59% can be reached. Obviously, with increasing the applied voltage, the fiber diameter decreased, while the mesh porosity increased gradually. The corresponding relationship between the fiber diameter and mesh porosity is shown in Figure 2.

Appendix A shows SEM images and fiber-diameter scatter diagrams of PEO/curdlan nanofiber films prepared by electrospinning under various DC voltages (12, 14, 16, 17, 17.5, 19, 22 and 24 kV). The polymer solution used in the electrospinning process was mixed with 6.0 wt.% PEO and 5.0 wt.% cardlan gum. Under all applied voltage conditions, the fiber diameter was lower than 100 nm. At the applied voltage of 12 kV, the lowest fiber diameter of 75 nm was obtained, while its mesh porosity was 20.99%. It can be seen that as the applied voltage was increased, the fiber diameter increased and the mesh porosity decreased (Figure 3). 

However, from SEM images shown in Appendix A, although the fibers are relatively fine under this condition, the bead-like fibers widely exist. In Appendix A, there are also many beaded fibers under low voltage conditions. However, the beaded fibers begin to decrease when the applied voltage is increased. Apparently, the appearance of bead-like fibers is dependent on the concentration of cardlan gum. That is, the higher the content of cardlan gum in the PEO, the easier it is to form the beaded nanofibers, and the harder it is to eliminate the beaded nanofibers via the increment of the applied voltage. 

Based on the above results, when the concentrations of curdlan gum in the solution are 1.0 and 5.0 wt.%, the fiber diameter increases gradually with increasing the operating voltage (Figure 1 and Figure 3). However, as observed in Figure 2, the fiber diameter decreases with the increase in the operating voltage when the concentration of curdlan gum in the solution is 2.0 wt.%. It can be seen that when the concentration of curdlan gum is 2.0 wt.%, the electrospinning solution is affected by high DC voltage, and the increase in electrostatic repulsion increases the stretching force of fiber filaments. This results in a thinner fiber diameter. Although the beaded fibers gradually disappear with the strengthening of the electric field when the concentration of curdlan gum is 1.0 wt.%, the fiber diameter continues to increase. 

Based on the above results, in order to avoid the appearance of beaded fibers in subsequent studies, the additional amount of 2.0 wt.% for the curdlan gum is more appropriate. When the concentration of curdlan gum is fixed at 2.0 wt.%, we can observe that the nanofiber diameter reduces to lower than 100 nm (93.51 nm) and the mesh porosity reaches 13.37% at the applied voltage of 19 kV. Although, when the applied voltage is increased to 22 and 24 kV, the fiber diameter can continue to decrease, there will occasionally be sharp noise and burnt smell occurring in the electrospinning system. Thus, to avoid short circuits in the electrospinning process (resulting in tiny sparks or charred areas on the surface of the collector plate or at the injection pump), the applied voltage of 19 kV is selected as the optimum condition. 

### 3.2. Effect of Working Distance on the Formation of Nanofiber Films

Theoretically, under sufficient operating voltage conditions, after elongating the working distance of electrospinning, the time for the electric field to act on the jet can be increased, and the stretching strength of fiber filaments is strengthened, which is beneficial to the reduction of fiber diameter. On the contrary, shortening the working distance will increase the electric field intensity, and the interaction between the electric field and jet will be accelerated, leading to a significant reduction in the fiber diameter. 

Immediately, to realize the effect of working distance on the formation of nanofiber films, various working distances (12, 13, 15, 17 and 20 cm) were used in the electrospinning process. Here, the operating voltage and feeding rate were fixed at 19 kV and 9 μL/min, respectively. Appendix A shows SEM images and fiber-diameter scatter diagrams of PEO/curdlan nanofiber films prepared by electrospinning under various working distances (12, 13, 15, 17 and 20 cm). The polymer solution used in the electrospinning process was mixed with 6.0 wt.% PEO and 1.0 wt.% cardlan gum. When the working distance was shorter than 15 cm, the nanofibers were mostly fine and uniform. However, as the working distance was longer than 15 cm, the average diameter of the nanofiber increased slightly to 130 nm, and the mesh porosity decreased significantly. It is obvious that the increment of the working distance has little effect on the fiber diameter, but the mesh porosity can decrease significantly with increasing the fiber diameter. The porosity is inversely proportional to the average nanofiber diameter, as shown in Figure 4. 

The increase in working distance can lead to a reduction in the fiber diameter since solvents in the longer distance have more time to evaporate. However, if the working distance is too long, the electric field strength acting on the jet will be insufficient, resulting in thicker filaments [19]. From the above result, when the concentration of cardlan gum in the PEO solution was only 1.0 wt.%, the electric field between the needle tip and collecting plate may be reduced if the working distance was more than 15 cm. Actually, the addition of cardlan gum also increased the conductivity. Therefore, it can be speculated that although the electrical conductivity of the solution had been improved after adding 1.0 wt.% cardlan gum, the generated electric field had not yet reduced the fiber diameter below 100 nm. As the working distance gradually increased, the electric field intensity was further reduced. This led to an increment in the fiber diameter. On the other hand, porosity refers to the ratio of the number of micropores per square millimeter of the film surface, so the decrease in porosity may also be related to the reduction in the number of filaments.

Appendix A shows SEM images and fiber-diameter scatter diagrams of PEO/curdlan nanofiber films prepared by electrospinning under various working distances (12, 13, 15, 17 and 20 cm). The polymer solution used in the electrospinning process was mixed with 6.0 wt.% PEO and 2.0 wt.% cardlan gum. Although the fiber diameter was generally still above 100 nm, even in the relatively short working distance (12 and 13 cm), the nanofibers were separated from each other. The sticking phenomenon occurred at the intersection of the nanofibers. In addition, as the working distance was increased, the fiber diameter also decreased. When the working distance was increased to 20 cm, the average fiber diameter even reached 99.2 nm. On the other hand, as the fiber diameter decreased, the mesh porosity increased. When the working distance was 20 cm, the mesh porosity was 10.35%, as shown in Figure 5. 

When the working distance was between 12 and 17 cm, the diameter increased with the increase in the working distance although the fiber diameter is still above 100 nm. However, the diameter has begun to decrease significantly with the increase in working distance, and even after the working distance reached 20 cm, the downward trend of the average fiber diameter became more obvious. It can be deduced that it provides sufficient conductivity to the solution after the content of curdlan gum in the solution increases to 2.0 wt.%. The high conductivity represents the sufficient charge density contained on the droplet at the tip of the needle, leading to the strong tensile force of the electric field. Under the appropriate working distance, it is helpful to decrease the fiber diameter. 

Appendix A shows SEM images and fiber-diameter scatter diagrams of PEO/curdlan nanofiber films prepared by electrospinning under various working distances (12, 13, 15, 17 and 20 cm). The polymer solution used in the electrospinning process was mixed with 6.0 wt.% PEO and 5.0 wt.% cardlan gum. Under various working distances, the average diameter of the nanofibers was all below 100 nm. When the working distance was 15 cm, the lowest diameter of 69.8 nm can be reached, and the mesh porosity also increased with decreasing the fiber diameter. However, when the working distance was increased to 17 cm, the fiber diameter slightly increased. The mesh porosity was still inversely proportional to the fiber diameter, as shown in Figure 6. Even the PEO/curdlan nanofibers prepared by using the 5.0 wt.% cardlan gum are relatively fine; the presence of bead-like or spun-cone fibers can always be seen on the fiber surface. 

From the experimental results of Figure 6, it can be seen more clearly that the higher the content of curdlan gum (5.0 wt.%), the more significant the effect on the fiber diameter. Although the variation in the average fiber diameter first decreases and then increases as the working distance becomes longer, the fiber diameter is still below 100 nm. It is speculated that when the content of cardlan gum in the solution increases to 5.0 wt.%, the electrical conductivity increases and the charge density of the solution is enhanced. This induces the improvement in the force of the electric field on the jet, and the Coulomb repulsion is strengthened. Thus, the fiber diameter can become relatively slender. However, the elongation of the working distance still slightly reduces the electric field strength acting on the jet, making the fiber filaments slightly thicker. 

Although a high concentration (5.0 wt.%) of curdlan gum can help to prepare the finer nanofibers, bead-like or spun-cone-like fibers still appear. Because the average fiber diameter is larger when the concentration of curdlan gum is 1.0 wt.%, the addition of 2.0 wt.% curdlan gum is selected as the optimum condition. On the other hand, based on the above results, when the concentration of curdlan gum is fixed at 2.0 wt.%, the working distance of 20 cm is more suitable to prepare the PEO/curdlan nanofibers. 

### 3.3. Effect of Polymer Solution Feeding Rate on the Formation of Nanofiber Films

When the reasonable operating voltage and working distance have been obtained, then it was necessary to find a suitable feeding rate of the polymer solution to help the formation of the Taylor cone on the nanofibers. Theoretically, the feeding rate is directly proportional to the formation amount of electrospun filaments, and it is also positively correlated with the thickness of filament diameter. 

Appendix A shows SEM images and fiber-diameter scatter diagrams of PEO/curdlan nanofiber films prepared by electrospinning under various feeding rates (5, 8, 9, 11, 15, 20, 25, 30, 35, 40 and 50 μL/min). Here, the operating voltage and working distance were fixed at 19 kV and 20 cm, respectively. The polymer solution used in the electrospinning process was mixed with 6.0 wt.% PEO and 1.0 wt.% cardlan gum. Although most of the nanofibers were formed in the fine-filament shape, no matter what the feeding rate was, the bead-like protrusions or spin-cone-like fibers would appear. When the feeding rate was 20 μL/min, a large number of spherical fibers appeared. As the feeding rate was increased to 35 μL/min, the shape of filaments could not be seen in the film. Under the feeding rate of 5~9 μL/min, the average fiber diameter increased to 131.8 nm, and the mesh porosity decreased to 10.22%. However, when the feeding rate was higher than 11 μL/min, the average fiber diameter reduced, and its value could drop to 86.1 nm (even without the formation of fiber filaments). Meanwhile, the mesh porosity reached a minimum value of 0.015% as the feeding rate was higher than 11 μL/min, as shown in Figure 7. 

From Appendix A, there are two problems with nanofiber films electrospun by PEO solution containing 1.0 wt.% curdlan gum at various feeding rates. One is that the filaments contain many beaded protrusions. The reason for the formation of beaded fibers is generally believed to be related to the concentration, viscosity, surface tension and conductivity of the solution. For example, when the viscosity of the solution is too low, it is easier to form beaded fibers. However, when the feeding rate is too high, due to the excessive amount of liquid flowing out of the needle tip, it exceeds the bearing capacity of the electric field tensile force generated under the fixed voltage. As a result, the jet cannot completely evaporate the solvent in the solution within the process time, causing the solvent to remain in the fiber. This makes the filaments quite wet, causing the fibers to stick to each other. 

The second problem is that the diameter of filaments increases first and then decreases as the feeding rate increases. When the feeding rate is lower than 9 μL/min, the fiber diameter increases with the increase in the feeding rate. However, the fiber diameter begins to decrease when the feeding rate is higher than 11 μL/min. We can observe the phenomenon that the number of spherical fibers increases a lot. The more beaded fibers, the faster the degradation rate. At the same time, the increase in the number of beaded fibers will also reduce the diameter of the fiber filaments, making it have a larger specific surface area and be more easily penetrated by water, which will accelerate the degradation rate. Therefore, we speculate that the reason for the decrease in the fiber diameter is related to the increase in the number of beaded fibers. 

Appendix A shows SEM images and fiber-diameter scatter diagrams of PEO/curdlan nanofiber films prepared by electrospinning under various feeding rates (5, 8, 9, 11, 15, 20, 25, 30, 35 and 40 μL/min). The polymer solution used in the electrospinning process was mixed with 6.0 wt.% PEO and 2.0 wt.% cardlan gum. When the feeding rate was 5–9 μL/min, there were no bead-like fibers formed in the nanofiber film, and the fibers were relatively thinner. The average fiber diameter was mostly below 100 nm (the smallest one can reach to 36.3 nm), and the mesh porosity was 65.29%. As the feeding rate continued to increase, the fiber diameter also increased. As the feeding rate was higher than 15 μL/min, not only beaded fibers but also a large number of plaques appeared in the filaments. As shown in Figure 8, with an increment in feeding rate, the fiber diameter also increased. However, the mesh porosity displayed a downward trend, and the diameter was inversely proportional to the porosity. 

In Appendix A, when the feeding rate is 15 µL/min, the nanofiber film has many beaded fibers and plaques, but the diameter of the nanofiber filaments still continues to increase. From this point of view, when the flow rate exceeds 15 µL/min, the amount of solution discharged from the needle tip exceeded the range that the electric field force can bear. This causes the solvent to be difficult to volatilize, making the fiber relatively moist. Thus, the beaded fibers and plaques formed in the nanofiber film. 

Appendix A shows SEM images and fiber-diameter scatter diagrams of PEO/curdlan nanofiber films prepared by electrospinning under various feeding rates (5, 9, 10, 15, 20, 25, 30, 35, 40, 45 and 50 μL/min). The polymer solution used in the electrospinning process was mixed with 6.0 wt.% PEO and 5.0 wt.% cardlan gum. Under various feeding rates, the average diameter of the nanofibers was mostly below 100 nm. Although the average diameter of nanofiber shows an increasing trend with the increase in feeding rate, the change is not obvious. At the same time, the mesh porosity is slightly increased with the increase in fiber diameter, but it is still relatively dense. As shown in Figure 9, the maximum porosity is only 13.31%. On the other hand, although there are not a large number of bead-shaped fibers or plaques in the nanofibers, there are occasional spin-cone fibers and the distribution of fiber diameters is uneven. 

In Appendix A, although the nanofiber films still have beaded protrusions, the fibers are relatively slender. In addition, the occurrence of spun tapered fibers is less, but there is still the problem of uneven thickness of fiber filaments. It is speculated that the formation of thinner fibers should be due to the higher content of curdlan gum, which increases the electrical conductivity. Thus, a larger electric field force is formed on the jet, and the fiber is stretched more completely, leading to finer fiber filaments. 

According to the above results, when the nanofiber film was prepared from the PEO solution containing 1.0 wt.% cardlan gum, the fiber filaments were relatively thicker under low feeding rates. When the flow rate was higher, although the fiber filaments were thinner, there were more bead-like fibers, and even the fibers stuck together. As the nanofiber film was prepared by using a PEO solution containing 5.0 wt.% cardlan gum, the fibers were relatively thinner, and there were fewer beaded fibers. However, the mesh porosity was too low in these nanofiber films, it may seriously affect the air permeability. Therefore, the films produced by using the cardlan gums with these two concentrations (1.0 and 5.0 wt.%) are not considered. 

Finally, considering the film prepared from the PEO solution containing 2.0 wt.% cardlan gum, it can be found that the fibers were thicker and there were more beaded fibers and plaques under high feeding rates. At low feeding rates, the fibers became relatively thin. When the feeding rate was 5 µL/min, the average fiber diameter was extremely low (only 36.3 nm), and the mesh porosity can reach 65.29%. However, it is difficult to maintain a stable and continuous supply of jet flow under a low feeding rate of 5 µL/min. Therefore, in this study, the feeding rate of 9 µL/min was used for the subsequent electrospinning experiments. 

### 3.4. Wetting and Disintegration Times of the PEO/Curdlan Nanofiber Films 

In this research, the purpose of fabricating the electrospun nanofibers mixed with PEO and cardlan gum is to prepare a fast-dissolving film that can quickly dissolve and release active ingredients through these materials, expanding its applications in facial masks and wound dressings. Therefore, it is important to understand the required time of wetting and disintegration for the nanofiber films in water and water vapor. In general, the nanofiber film prepared by electrospinning has a high specific surface area and is a highly porous structure, so most of the electrospun nanofibers films can dissolve/disintegrate quickly to deliver the desired drug. 

Based on the above results, the PEO/curdlan nanofiber films were prepared by using 6.0 wt.% PEO solution containing 2.0 wt.% cardlan gum. Additionally, the 5.0 wt.% quercetin inclusion complex was also mixed in PEO/curdlan nanofiber films for wetting and disintegration processes. Quercetin widely exists in many plants, flowers, leaves and fruits. In terms of medicinal effects, quercetin has excellent cough-relieving and phlegm-relieving effects [20,21]. Additionally, it also has considerable effects on anti-inflammatory, anti-allergic, anti-tumor, anti-oxidation and lowering blood pressure. Moreover, the operating voltage, working distance and feeding rate of the polymer solution for the electrospinning process were 19 kV, 20 cm and 9 µL/min, respectively. Figure 10 shows the SEM image of the quercetin blended with the PEO/curdlan nanofiber film. 

Figure 11 shows the photographs of the electrospun nanofiber instant film after performing the wetting process on the wet wipe for 5 min. It can be observed that the instant film was obviously dissolved within 15 s. At the same time, the quercetin inclusion complex in the instant film was adsorbed on the wet wipe. In this experiment, the amount of used water was sparse, and it mainly simulated the moisture condition of the skin after washing the face or applying the toner. The time required for moisture to penetrate the film in the experiment can reflect the time required for the instant film to release the active ingredient under the condition of low water content on the skin surface. 

Figure 12 shows the photographs of the electrospun nanofiber instant film disintegrated in water. It can be seen that the film disintegrated instantly when it touched water, and the disintegration time was only about 4~5 s. When the amount of water is sufficient, the film is completely dissolved, and the quercetin inclusion complex also can be dissolved in water. 

Figure 13 shows the photographs of the electrospun nanofiber instant film wetted and dissolved by 50 °C water vapor for 30 min. We can find that the film began to be transparent within 15 s when it encountered water vapor. After immersing the instant film in water vapor for 135 s, it became more and more transparent. After 165 s, tiny holes appeared on the instant film, and there was also a clear shrinkage change in the film. After 225 s, the hole in the film gradually expanded. At 255 s, there were obvious holes in the film. At 405 s, the film has been significantly dissolved. Finally, when the instant film was immersed in water vapor for 30 min, it almost completely disintegrated. 

According to the wetting and disintegration experimental results, it can be known that the electrospun PEO/curdlan nanofiber film prepared in this study is very soluble in water and releases active ingredients quickly. Obviously, it has great application potential in instant masks and quick-release wound dressings. In addition, this film also has promising applications in environments with sufficient water vapor (such as bathhouses, saunas, and hot spring clubs, etc.). 

Compared with previous studies, our electrospun PEO/curdlan nanofiber films are significantly faster in dissolving in water and releasing active ingredients. Especially the disintegration time (in water) of only 4~5 s shows that our PEO/curdlan nanofiber films have a very high potential for biomedical applications. 

The nanofiber film prepared by electrospinning technology has a large specific surface area, ductility, adjustable porosity, excellent moisture retention, absorption and conformability, and can also carry drugs or other functional ingredients. It can be used as a wound dressing and added biologically active substances such as growth factors, antibacterial ingredients, etc., to prevent wound infection. At the same time, due to its biodegradable characteristics, it can gradually disintegrate during the wound repair process, avoiding the disadvantages of traditional gauze that would cause discomfort to the affected part during the removal process due to sticking tissue. In addition, compared with traditional masks, the essential components of biocompatible polymer-soluble nanofiber masks are less likely to evaporate and dry out. Meanwhile, due to its dissolvable property, it can effectively reduce the subsequent environmental issues derived from waste masks. 

## 4. Conclusions 

In summary, for the preparation of high-quality electrospun PEO/curdlan nanofiber films, the 6.0 wt.% PEO solution mixed with the 2.0 wt.% curdlan gum was used. Moreover, when the electrospinning process was performed with the optimum conditions including the operating voltage of 19 kV, the working distance of 20 cm and the polymer-solution feeding rate of 9 μL/min, thinner nanofibers with a higher mesh porosity (without the formation of beaded fibers) can be obtained in the PEO/curdlan nanofiber film. The PEO/curdlan nanofiber instant film (mixed with 5.0 wt.% quercetin inclusion complex) was dissolved on the wet wipe within 15 s, revealing it can be quickly dissolved in the skin surface after washing the face or applying toner. When the instant film encountered water, it can disintegrate instantly. This indicates that the film can be completely dissolved in a well-hydrated situation, resulting in an efficient dissolution of the quercetin inclusion complex. When the instant film was wetted with water vapor at 50 °C, it became transparent within 15 s and obvious holes were formed in the film after 250 s. After 30 min, the instant film was almost completely disintegrated in the water vapor. The results prove that the electrospun PEO/curdlan nanofiber film has a high potential in biomedical applications, especially for instant masks and quick-release wound dressing. The nanofiber films prepared in this study have advantages including fine diameter, large specific surface area, mesh-porous structure, high porosity and light weight. At the same time, this nanofiber film is highly soluble in water and releases active ingredients quickly. However, how to effectively mass-produce the electrospinning process and reduce the process cost are the problems to be solved. 

## Figures and Tables

**Figure 1 materials-16-03863-f001:**
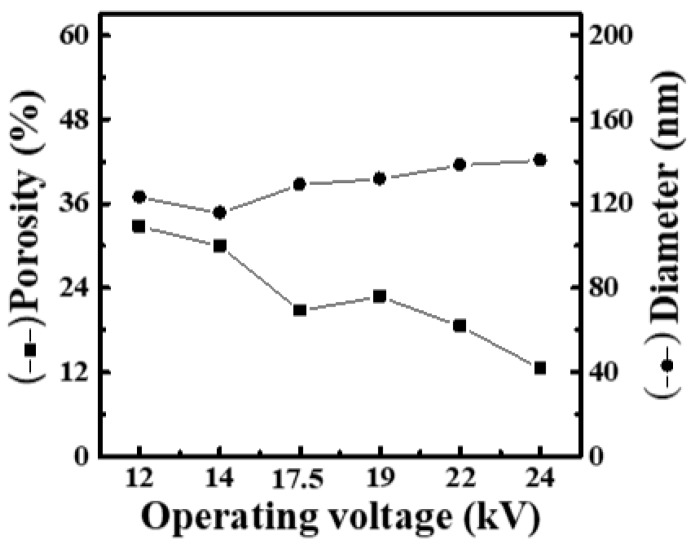
Relationship between fiber diameter and mesh porosity of electrospun PEO/curdlan nanofibers fabricated at various operating voltages. The concentrations of PEO and curdlan gum are 6.0 and 1.0 wt.%, respectively.

**Figure 2 materials-16-03863-f002:**
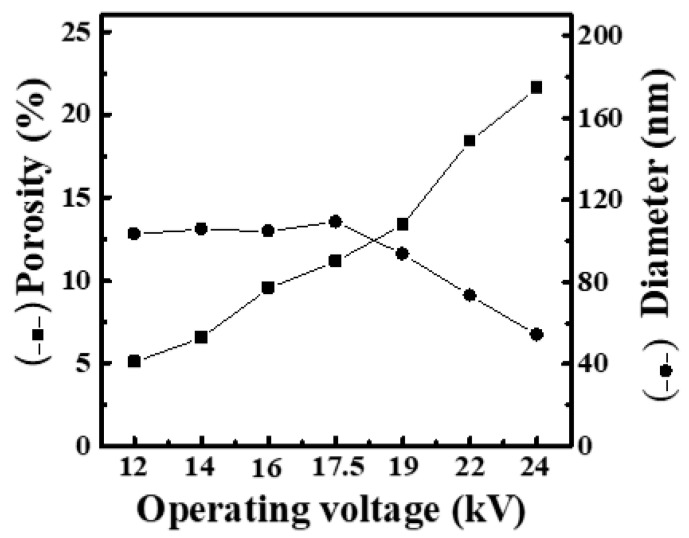
Relationship between fiber diameter and mesh porosity of electrospun PEO/curdlan nanofibers fabricated at various operating voltages. The concentrations of PEO and curdlan gum are 6.0 and 2.0 wt.%, respectively.

**Figure 3 materials-16-03863-f003:**
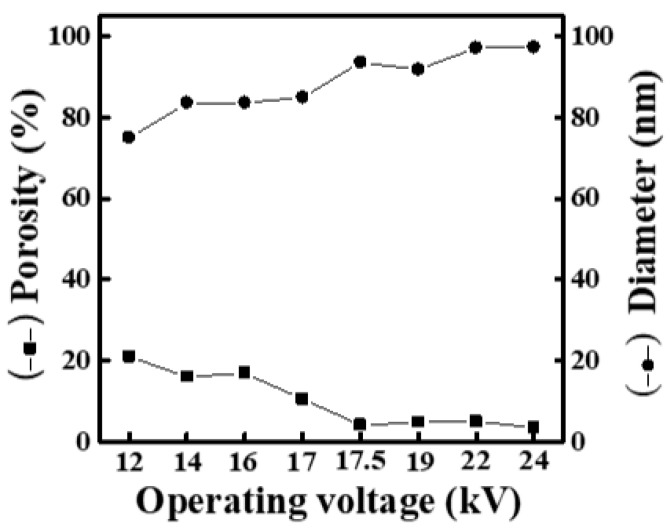
Relationship between fiber diameter and mesh porosity of electrospun PEO/curdlan nanofibers fabricated at various operating voltages. The concentrations of PEO and curdlan gum are 6.0 and 5.0 wt.%, respectively.

**Figure 4 materials-16-03863-f004:**
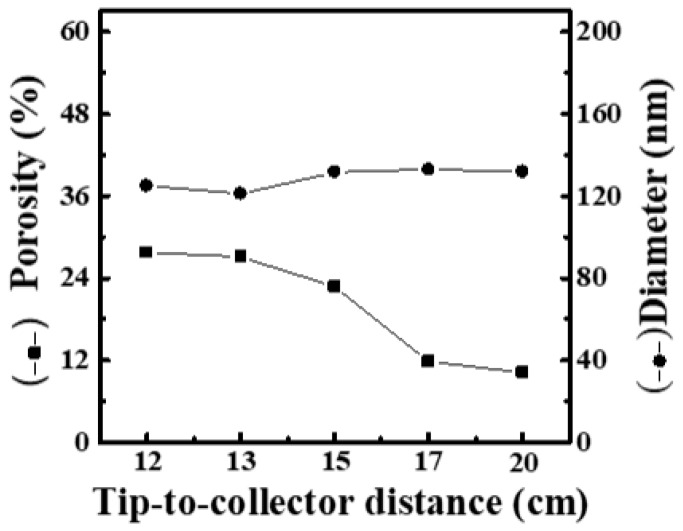
Relationship between fiber diameter and mesh porosity of electrospun PEO/curdlan nanofibers fabricated at various working distances. The concentrations of PEO and curdlan gum are 6.0 and 1.0 wt.%, respectively.

**Figure 5 materials-16-03863-f005:**
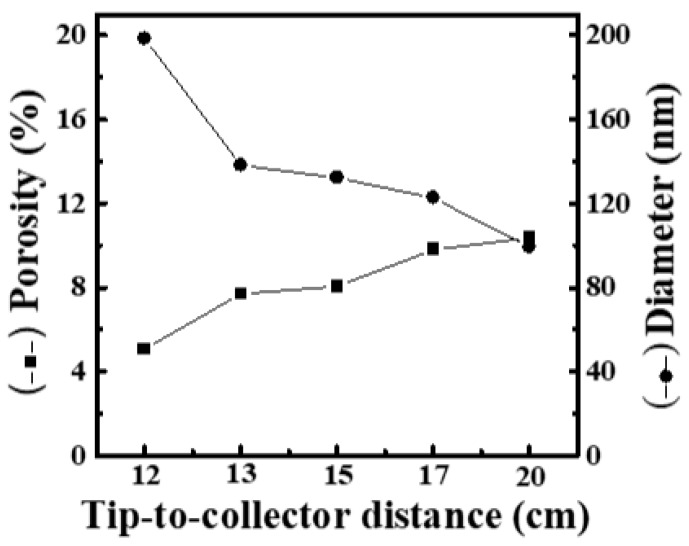
Relationship between fiber diameter and mesh porosity of electrospun PEO/curdlan nanofibers fabricated at various working distances. The concentrations of PEO and curdlan gum are 6.0 and 2.0 wt.%, respectively.

**Figure 6 materials-16-03863-f006:**
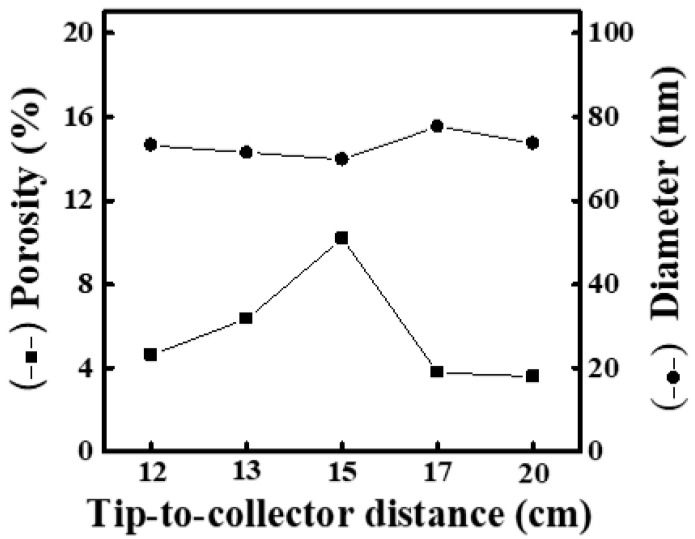
Relationship between fiber diameter and mesh porosity of electrospun PEO/curdlan nanofibers fabricated at various working distances. The concentrations of PEO and curdlan gum are 6.0 and 5.0 wt.%, respectively.

**Figure 7 materials-16-03863-f007:**
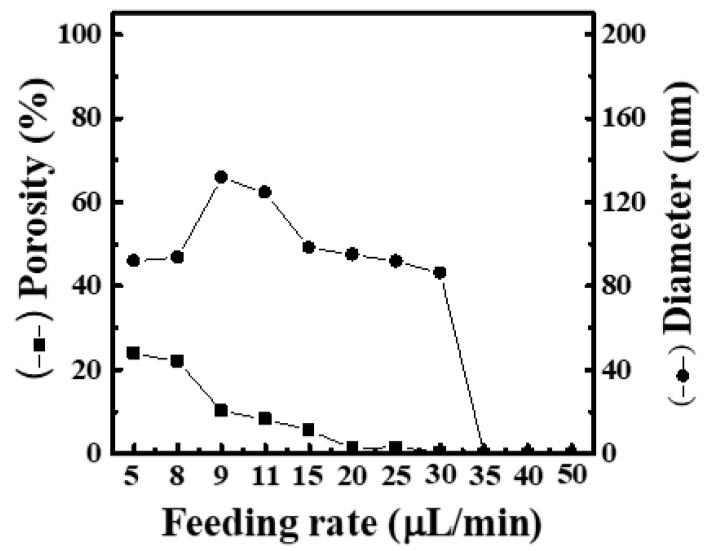
Relationship between fiber diameter and mesh porosity of electrospun PEO/curdlan nanofibers fabricated at various feeding rates. The concentrations of PEO and curdlan gum are 6.0 and 1.0 wt.%, respectively.

**Figure 8 materials-16-03863-f008:**
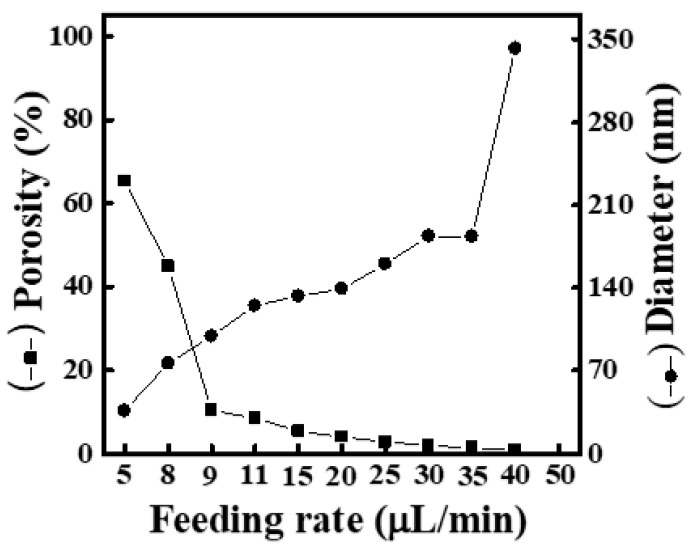
Relationship between fiber diameter and mesh porosity of electrospun PEO/curdlan nanofibers fabricated at various feeding rates. The concentrations of PEO and curdlan gum are 6.0 and 2.0 wt.%, respectively.

**Figure 9 materials-16-03863-f009:**
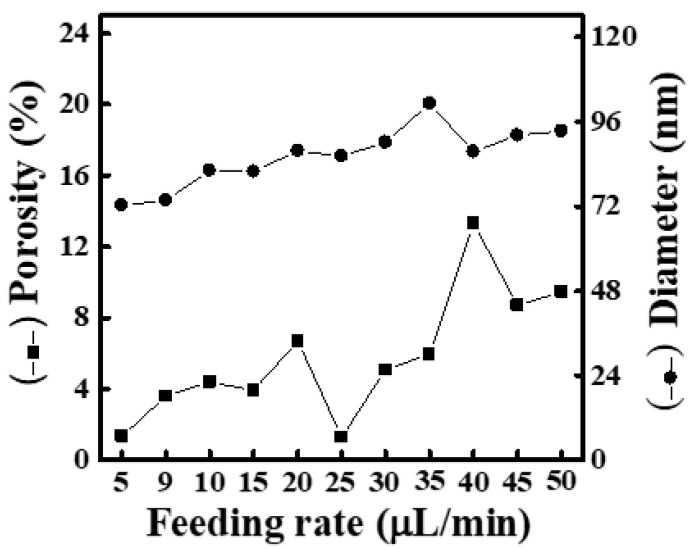
Relationship between fiber diameter and mesh porosity of electrospun PEO/curdlan nanofibers fabricated at various feeding rates. The concentrations of PEO and curdlan gum are 6.0 and 5.0 wt.%, respectively.

**Figure 10 materials-16-03863-f010:**
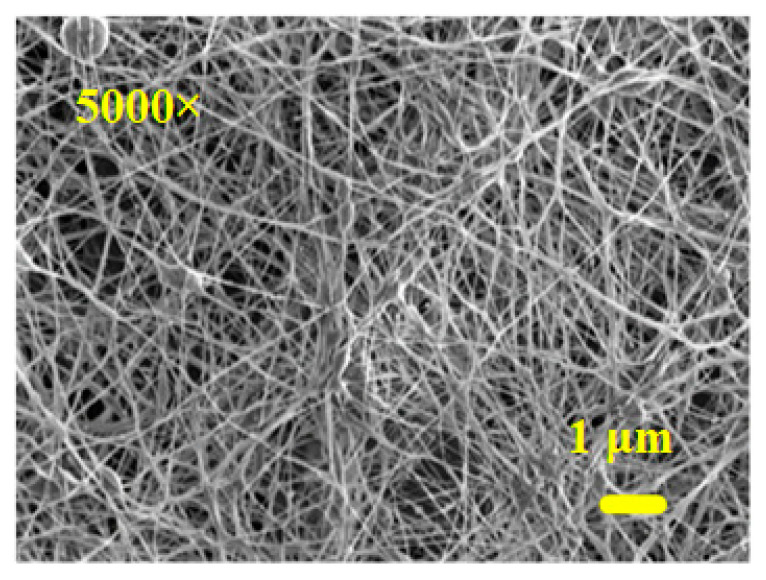
SEM image of the quercetin blended with the PEO/curdlan nanofiber film.

**Figure 11 materials-16-03863-f011:**
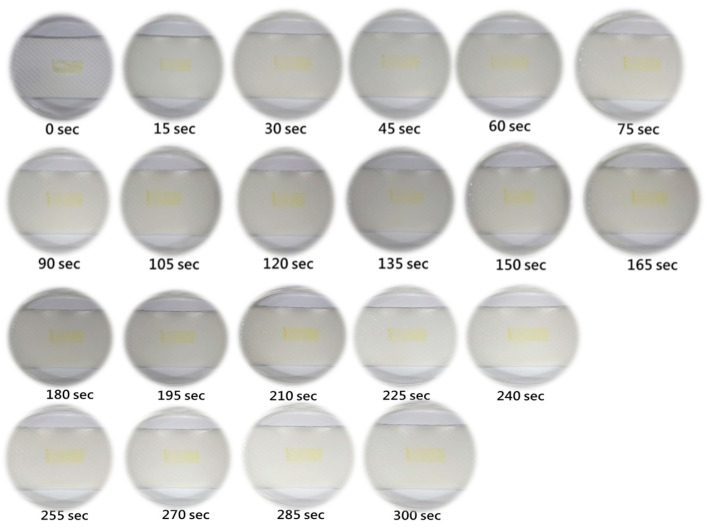
Photographs of the electrospun nanofiber instant film after performing the wetting process on the wet wipe for 5 min.

**Figure 12 materials-16-03863-f012:**
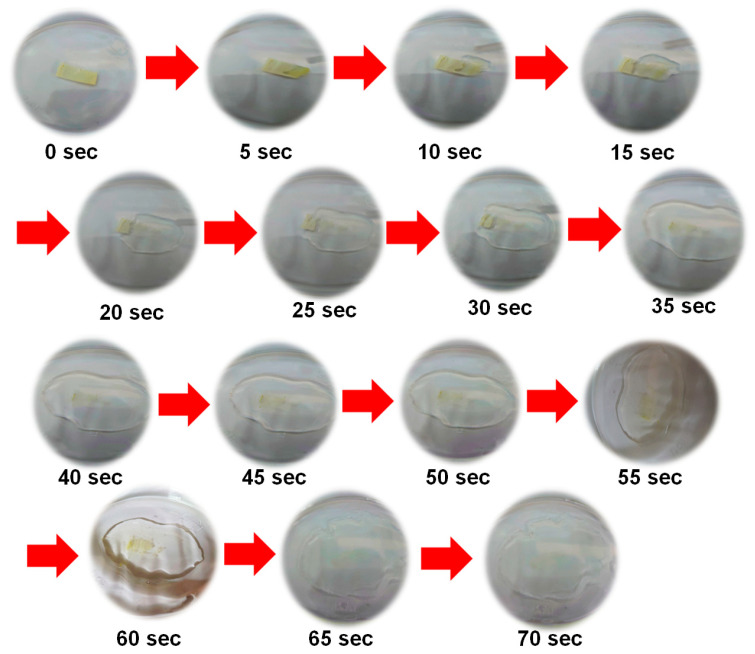
Photographs of the electrospun nanofiber instant film disintegrated in water.

**Figure 13 materials-16-03863-f013:**
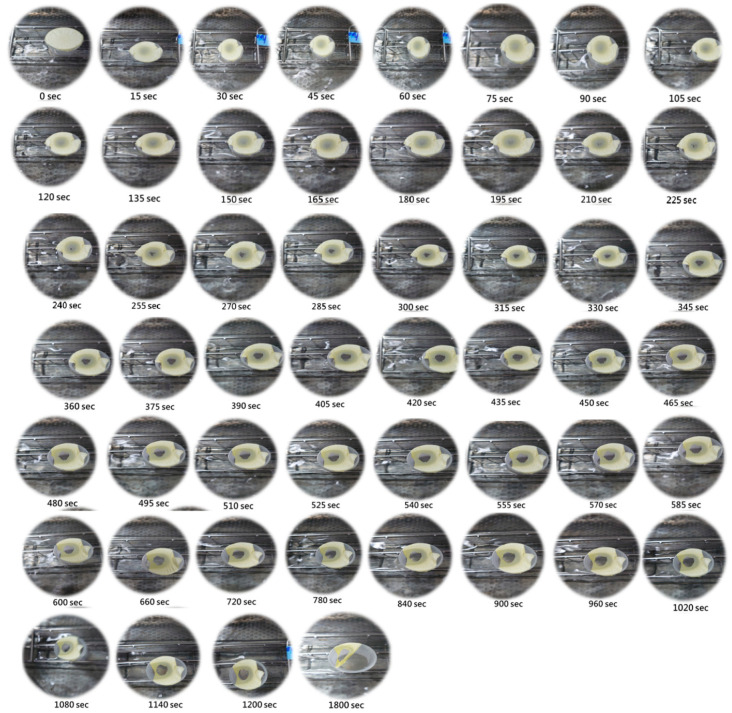
Photographs of the electrospun nanofiber instant film wetted and dissolved by 50 °C water vapor for 30 min.

## Data Availability

Data is unavailable due to privacy or ethical restrictions.

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
