# Peer review of "Preparation and Characteristics of Polyethylene Oxide/Curdlan Nanofiber Films by Electrospinning for Biomedical Applications"

_materials, 2023, doi:10.3390/ma16103863_

Round 1

Reviewer 1 Report

This manuscript studies the electrospinning process of polyethylene oxide/curdlan nanofiber films. The subject is interesting; however, some revisions need to be addressed before final decision.

1. It is strongly recommended to avoid using the lumped references such as [1-6]. It is acceptable in s research article.

2. There are 13 references in two lines of Line 62 and Line 63. This is not acceptable for a research paper. Any reference needs to be discussed and justified.

3. The literature survey is weak. The authors should conduct an extensive literature review on the recent publications in the field to highlight the novelties of this study.

4. Please clearly highlight the novelties of the study in the last paragraph of the Introduction. What is the new concept in this study? The materials, the methodology, the composition, or etc.? What did this study add to the literature?

5. In the methodology, please clearly state why this range is considered for the parameters?

6. What is the basis to select these values for the levels of the parameters? For instance, the voltage has been considered in these levels, 12-14-17.5-19-22-24, with the intervals of 2-3.5-1.5-3-2. Generally, it is better to consider the levels with the same intervals. This issue is valid for other parameter and levels.

7. The results have been only reported without any discussion. For instance, for Figure 4, it is stated that “Obviously, with increasing the applied voltage, the fiber diameter was decreased, while the surface porosity was increased gradually.” Yes, it is too obvious. Anyone looking at the figure can realize this fact. However, the authors should discuss the results in detail and state why an output decreases or increases with satisfactory physical reasons.

8. There are too many figures. Some of them can be transferred to the supplementary information. For instance, Figures 3, 5, 7, 9, 11, 13, 15, and 17 can be transferred to the supplementary information because their main results, i.e. porosity and diameter, have been presented in other figures.

Author Response

We appreciate the comments and suggestions from the reviewer 1, and the manuscript has been modified. In the revised manuscript, the added and modified texts are shown in red words. The detailed changes are listed as follows.

Reviewer 2 Report

This study investigated the properties of PEO/curdlan nanofiber films produced by electrospinning and the effect of operating parameters, including operating voltage, working distance, and feeding rate of polymer solution, on the film's properties. The feasibility of PEO/curdlan nanofiber films for biomedical applications has been assessed through wetting and disintegration processes. The topic of the study is interesting, but the manuscript needs major revision to be publishable in the Materials Journal.

 -    While the introduction provides a comprehensive overview of the applications and properties of electrospun nanofiber films, I would appreciate it if the authors could clarify the specific novelty of their study in comparison to existing literature. Can they please highlight the unique contribution of their work and how it advances the current state of research in this field?

 -        The Material section is missing in the manuscript

 -       Experimental section needs to be completed; Which temperature has been used for solution preparation? What is the 90:10 ratio? The type of SEM is not mentioned, the method for fiber diameter measurement is missing, and the method of surface porosity measurement is missing

 -         The quality of SEM images is too low, and the labels are unclear!

  -          There is no porosity on the surface of the fibers! What does surface porosity mean? Does it mean the porosity of fibrous meshes?

 -          Usually, increasing distance leads to a decrease in fiber diameter since solvents in the higher distance have more time to evaporate. I would request that authors describe why in the case of 1% curdlan gum, increasing the distance had no significant effect on fiber diameter but porosity.

  -          I would request that authors describe how the presence of cardlan gum affects the solution properties, including viscosity and conductivity.

 -          I would request that authors describe how 5.0 wt.% quercetin inclusion complex was mixed within the PEO/curdlan nanofiber films. It is better that authors provide an SEM image of fiber mesh in the presence of 5.0 wt.% quercetin inclusion complex.

 -          Why did the authors not use a reference like PEO/curdlan cast film in wetting and disintegration times asses?

I would suggest that the authors work on improving the English language of the manuscript

Author Response

We appreciate the comments and suggestions from the reviewer 2, and the manuscript has been modified. In the revised manuscript, the added and modified texts are shown in red words. The detailed changes are listed as follows. 

Reviewer 3 Report

In this study, the polyethylene oxide (PEO) and curdlan solutions were used to prepare the PEO/curdlan nanofiber films by electrospinning. The results indicated that the PEO/curdlan nanofiber film is feasible for biomedical applications. The paper could be accepted after revision.

-PEO was used as the base material, and its concentration was fixed at 6.0 wt.%. Why the concentration was selected ?

-Solvent should be mentioned in the abstract.

-Suppliers of the materials should be described.

-It should be explained why curdlan is used here as biocompatible polymer materials?

-Chemical structures of the polymers should be demonstrated.

-Advantages and disadvantages of the nanofiber films should be presented in conclusions.

- Could properties of the nanofiber films compared with those of some commercial materials.

- Could properties of the nanofiber films compared with those of described in literature for biomedical applications?

Author Response

We appreciate the comments and suggestions from the reviewer 3, and the manuscript has been modified. In the revised manuscript, the added and modified texts are shown in red words. The detailed changes are listed as follows.

Round 2

Reviewer 1 Report

The revised manuscript is satisfactory. It is now acceptable.

Author Response

We sincerely appreciate this comment from the reviewer. 

Reviewer 2 Report

Dear Authors,

Thank you for providing the modified version of the manuscript. The only thing I would recommend to the authors is referring to the specific concentration of curdlan gum that has been used to achieve the 90:10 weight ratio with PEO.   

Minor editing of English language required

Author Response

We appreciate the comment and suggestion from the reviewer 2, and the manuscript has been modified. In the revised manuscript, the added and modified texts are shown in red words. The detailed changes are listed as follows.

Reviewer 3 Report

Accept in present form

Author Response

(The authors gave the same response as above.)
